# Iterative Reasoning Preference Optimization

**Richard Yuanzhe Pang**[1,2]    **Weizhe Yuan**[1,2]    **Kyunghyun Cho**[2]
**He He**[2]    **Sainbayar Sukhbaatar**[1*]    **Jason Weston**[1,2*]

[1]Meta FAIR          [2]New York University

## Abstract

Iterative preference optimization methods have recently been shown to perform well for general instruction tuning tasks, but typically make little improvement on reasoning tasks [Yuan et al., 2024, Chen et al., 2024]. In this work we develop an iterative approach that optimizes the preference between competing generated Chain-of-Thought (CoT) candidates by optimizing for winning vs. losing reasoning steps. We train using a modified DPO loss [Rafailov et al., 2023] with an additional negative log-likelihood term, which we find to be crucial. We show reasoning improves across repeated iterations of this scheme. While only relying on examples in the training set, our approach results in increasing accuracy on GSM8K, MATH, and ARC-Challenge for Llama-2-70B-Chat, outperforming other Llama-2-based models not relying on additionally sourced datasets. For example, we see a large improvement from 55.6% to 81.6% on GSM8K and an accuracy of 88.7% with majority voting out of 32 samples.

## 1   Introduction

Preference optimization has proven to give large gains when aligning pre-trained language models to human requirements compared to supervised fine-tuning alone [Ziegler et al., 2019, Stiennon et al., 2020]. Offline methods such as DPO [Rafailov et al., 2023] are becoming more popular for their simplicity and efficiency. Recent results have shown that iterative application of such an offline procedure is beneficial, whereby the updated model is used to construct new preference relations that are more informative, and hence improve results further. These methods include Iterative DPO [Xu et al., 2023, Xiong et al., 2023], Self-Rewarding LLMs [Yuan et al., 2024], SPIN [Chen et al., 2024], and other methods [Rosset et al., 2024]. Common to these approaches is that they have been shown to perform well on general instruction tuning tasks, but they either make only moderate gains or even decrease the performance on standard reasoning tasks. While other kinds of iterative training methods have been applied successfully to reasoning, particularly involving the iteration of supervised fine-tuning (SFT) such as STaR [Zelikman et al., 2022], Rest$^{EM}$ [Singh et al., 2024], and V-STaR [Hosseini et al., 2024][1], using preference optimization to train the generative reasoning model is not applied in these methods.

In this work, we develop an approach to apply iterative preference optimization to reasoning tasks, with a particular focus on Chain-of-Thought (CoT) reasoning [Wu et al., 2023]. On each iteration we sample multiple chain-of-thought reasoning steps and final answers over training prompts, and then construct preference pairs such that pair winners have correct answers and pair losers have wrong answers. We then train a variant of DPO that includes a negative log-likelihood (NLL) loss term for the pair winners, which also proves crucial for performance. Given the newly trained model, we then iterate the procedure by generating new pairs, and training again, starting from the previously trained

---

[*] Equal contribution.

[1]V-STaR does use preference optimization, but for training a separate verifier model.

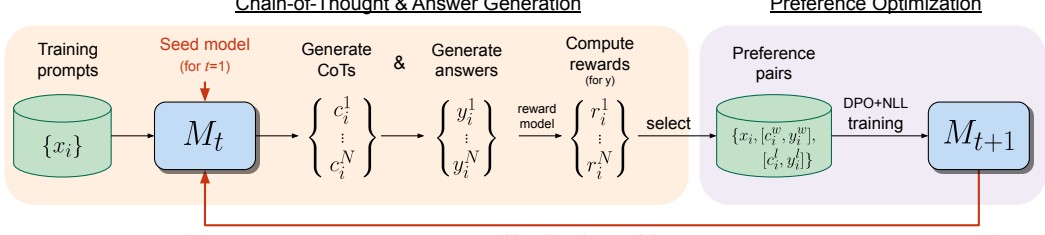

Figure 1: **Iterative Reasoning Preference Optimization.** Our iterative preference optimization method consists of two steps: (i) *Chain-of-Thought & Answer Generation*: training prompts are used to generate candidate reasoning steps and answers from model $M_t$, and then the answers are evaluated for correctness by a given reward model. (ii) *Preference Optimization*: preference pairs are selected from the generated data, which are used for training via a DPO+NLL objective, resulting in model $M_{t+1}$. This whole procedure is then iterated resulting in improved reasoning ability on the next iteration, until performance saturates.

model. We find that reasoning performance improves over multiple iterations until it eventually saturates.

We show that our approach, termed *Iterative Reasoning Preference Optimization* (Iterative RPO), outperforms a number of baselines, including SFT or applying standard DPO, as well as other baselines from the literature. We see an improvement from 55.6% of zero-shot performance on GSM8K to 81.6% after our Iterative RPO training (or from 70.7% to 88.7% with majority voting out of 32 samples), from 77.8% to 86.7% on ARC-Challenge (without using the provided ARC Corpus), and from 12.5% to 20.8% on MATH (or from 18.8% to 29.1% with majority voting out of 32 samples), without using the provided pretraining corpus in MATH. We provide ablations that indicate the components that lead to these improvements. We also present analysis on how different objectives influence the probabilities of training sequences, which helps explain the success of our method. Overall, our method provides a simple recipe that has the potential to improve the reasoning ability of LLMs over a wide range of tasks, as shown on the three tasks we consider.

## 2   Iterative Reasoning Preference Optimization

Our approach first assumes access to a base, typically pretrained or instruction-tuned, language model, a set of training inputs, and the ability to judge the correctness of the final outputs. Given a training input, the language model is expected to generate (i) a set of reasoning steps (Chain-of-Thought), followed by (ii) a final answer to the given problem. We assume that we have access to a correctness measure for the final answer, and not for the correctness of the reasoning steps used to reach that answer. In our experiments, we thus consider datasets where gold labels are provided for training inputs, and a binary reward is derived by the exact match between these labels and the final answer generations. However, our approach could also be applied to settings with more general reward models.

On each iteration, our method consists of two steps, (i) Chain-of-Thought & Answer Generation and (ii) Preference Optimization, as shown in Figure 1. For the $t^{\text{th}}$ iteration, we use the current model $M_t$ in step (i) to generate new data for training the next iteration's model $M_{t+1}$ in step (ii).

**Initialization.**   We assume we are given an initial model $M_0$, and a training set $D = \{(x_i, y_i)\}_i$ containing questions $x_i$ and their correct answers $y_i$. The model will be trained and updated at each iteration, resulting in models $M_0, M_1, \ldots, M_T$.

**Chain-of-thought & answer generation.**   Given the current model $M_t$, we generate $N$ different responses for every input, where each response consists of CoT reasoning $c$ followed by a final answer $y$:

$$(c_i^n, y_i^n) \sim M_t(x_i) \quad \text{for all } x_i \in D \text{ and } n \in [N],$$

where we use $[N]$ to denote $\{1, 2, \ldots, N\}$.

In the general version of our approach, one then computes the reward $r_i^n$ for each of these responses based on the correctness of their answers, i.e., $r_i^n = R(y_i^n, y_i)$. In our experiments this simply corresponds to $r_i^n = 1$ if $y_i^n = y_i$, and 0 otherwise; i.e., whether the prediction matches the answer provided in the training dataset. Thus we have constructed a set of generated responses augmented with rewards:

$$G_i = \{c_i^n, y_i^n, r_i^n\}_{n \in [N]}.$$

**Preference optimization.** In the next step, we first construct a dataset of response pairs $D_t^{\text{pairs}}$ based on the generations $G_i$ from the current model $M_t$. The paired data is constructed such that chosen (winning) responses have higher rewards than rejected (losing) responses. This data is then used for preference optimization. In general, this can be done by selecting two responses for the same input, such that one has higher reward than the other, and setting the one with higher reward as the winner. In the binary reward case, we can split the generated responses $G_i$ into two sets based on their rewards:

$$G_i^w = \{c_i^n, y_i^n \mid r_i^n = 1\},$$
$$G_i^l = \{c_i^n, y_i^n \mid r_i^n = 0\}.$$

Next we build a dataset of preference pairs by selecting a winner response $(c_i^w, y_i^w)$ from $G_i^w$, and a loser response $(c_i^l, y_i^l)$ from $G_i^l$. In particular, we simply iterate over $G_i^w$ and $G_i^l$ simultaneously[2] to produce $K$ pairs of indices $\{(w_k, l_k)\}$, in order to ensure we use as much of the data as possible.

$$D_t^{\text{pairs}} = \{(c_i^{w_k}, y_i^{w_k}), (c_i^{l_k}, y_i^{l_k}) \mid x_i \in D \text{ and } k \in [K]\}.$$

Given the preference pairs, we can now train a new model $M_\theta$ that will become our next model $M_{t+1}$. The parameters $\theta$ are initialized from model $M_t$, and updated with a loss function that combines the DPO loss [Rafailov et al., 2023] for learning from the preference pairs, and the negative log-likelihood (NLL) loss for learning over the winning response from each pair. The loss corresponding to each preference pair is as follows:

$$\begin{aligned}
\mathcal{L}_{\text{DPO+NLL}} &= \mathcal{L}_{\text{DPO}}(c_i^w, y_i^w, c_i^l, y_i^l | x_i) + \alpha \mathcal{L}_{\text{NLL}}(c_i^w, y_i^w | x_i) \\
&= -\log \sigma \left( \beta \log \frac{M_\theta(c_i^w, y_i^w | x_i)}{M_t(c_i^w, y_i^w | x_i)} - \beta \log \frac{M_\theta(c_i^l, y_i^l | x_i)}{M_t(c_i^l, y_i^l | x_i)} \right) - \alpha \frac{\log M_\theta(c_i^w, y_i^w | x_i)}{|c_i^w| + |y_i^w|}. \quad (1)
\end{aligned}$$

Here $M(x)$ denotes the probability of sequence $x$ under the model $M$, and $\sigma$ is the sigmoid function. We use the previous iteration's model $M_t$ as the reference model in the denominator of the DPO term. Note that the NLL term is normalized by the total response length. The hyperparameter $\alpha$ balances the two loss terms. For brevity we omit the pair index $k$, but we optimize this loss on each of the $k \in [K]$ pairs generated for every input sample. At the end of this training, we thus obtain our next model $M_{t+1} = M_\theta$, which will be then used to build data for the subsequent iteration.

**Iterative training.** Our overall procedure trains a series of models $M_1, \ldots, M_T$ where each successive model $t + 1$ uses preference data $D_t^{\text{pairs}}$ created by the $t^{\text{th}}$ model.

In our experiments, we define the models and the training data they use as follows:

$M_0$ : Base LLM; in our experiments we initialize with a fine-tuned instruction following model.

$M_1$ : Initialized with $M_0$, then trained with $D_0^{\text{pairs}}$ using $\mathcal{L}_{\text{DPO+NLL}}$.

$M_2$ : Initialized with $M_1$, then trained with $D_1^{\text{pairs}}$ using $\mathcal{L}_{\text{DPO+NLL}}$.

$M_3$ : Initialized with $M_2$, then trained with $D_2^{\text{pairs}}$ using $\mathcal{L}_{\text{DPO+NLL}}$.

$M_4$ : Initialized with $M_3$, then trained with $D_3^{\text{pairs}}$ using $\mathcal{L}_{\text{DPO+NLL}}$.

This approach can be seen as a similar, but simpler, instance of the Self-Rewarding LLM training scheme proposed in Yuan et al. [2024], with three differences. *Firstly*, on each iteration in Self-Rewarding a new set of prompts is created to explore the input distribution, but in our approach we use the same fixed set of prompts. *Secondly*, due to this choice our experimental setup does

---

[2]If the iteration reaches the end of a set, it restarts from the first element. If one of the sets is empty, then that input will be ignored.

not require a sophisticated reward model to judge the model generations, as we assume the training prompts have provided gold labels which we compare to. These two omitted steps are challenging for reasoning tasks because they require a language model to verify correctness, which is known to be difficult [Huang et al., 2024]. *Thirdly*, we show that our DPO+NLL objective is important for our reasoning tasks, whereas Self-Rewarding LLM has used the standard DPO objective.

Our approach is also related to the iterative training in the Self-Taught Reasoning (STaR) method [Zelikman et al., 2022], except that their approach uses SFT training, rather than preference optimization using DPO-like training. Preference optimization allows the use of negative examples of reasoning chains and answers, which we show improves performance. See Section 4 for more discussion of related work.

## 3 Experiments

### 3.1 Math Word Problems: GSM8K

In our first set of experiments, we use the GSM8K dataset [Cobbe et al., 2021][3] that contains real grade-school math word problems. For example the question: *"Natalia sold clips to 48 of her friends in April, and then she sold half as many clips in May. How many clips did Natalia sell altogether in April and May?"*. These questions typically require the model to perform intermediate reasoning, i.e., generating chain-of-thought before answering, otherwise performance is poor. Each problem contains a question $x_i$, gold chain-of-thought solution $c_i$, and a final numerical answer $y_i$. For our entire training process, we only use the training set of around 7.5k problems without any extra questions.

**Experimental setup.** As a seed model $M_0$ we use the chat version of Llama-2 70B model [Touvron et al., 2023], which is instruction fine-tuned. We use a zero-shot prompt containing the question together with instructions to produce a chain-of-thought and to follow a specific format so the final answer can be easily extracted (the exact prompt is given in Appendix B.2). In each iteration, we generate $N = 30$ solutions per problem using sampling with temperature 0.8 for iterations 1–2 and temperature 1.3 for iterations 3–4 (hoping that there is a significant number of incorrect generations in later iterations). Since some problems might not have any model-generated correct solution, we include the gold human written solution $(c_i, y_i)$ in the winning set $G_i^w$ so it is not empty. Then we generate $K = 10$ pairs per problem for training with our loss in Equation 1, and filter out examples that were too long in terms of overflowing the context length or else do not have any incorrect generations. This procedure gives around 55–60k pairs for training, per iteration.[4]

In total, we perform four iterations, producing models $M_1$, $M_2$, $M_3$, and $M_4$. For each iteration, we train a maximum of 5000 steps, and then select the best checkpoint using a held-out 1k samples from the training set. We then retrain while including those 1k samples for the selected number of steps. The coefficient $\alpha$ is tuned in $\{0.25, 0.5, 1, 2\}$ when training $M_1$, and we end up using 1 for all experiments in the paper. The coefficient $\beta$ in the DPO loss is tuned in $\{0.05, 0.1, 0.5, 1.0\}$, and we end up using 0.1 in this experiment. We use a batch size of 16 and a learning rate 7e-7 using the AdamW optimizer. Throughout this paper, all generation is done using one node containing eight V100 GPUs (32G memory). To do inference efficiently, we use vLLM [Kwon et al., 2023]. All training is done using eight nodes each containing eight A100 GPUs (80G memory).

Overall results are given in Table 1, where we give the exact match accuracy on the GSM8K test set.

**Iterative RPO improves over baselines.** We find that Iterative RPO outperforms zero-shot CoT, supervised fine-tuning (SFT) on the gold (dataset-provided) CoT solutions, and variants of DPO by a wide margin. SFT gives a boost in performance compared to zero-shot CoT from 55.6% to 63.5% but still far from the 81.6% of Iterative RPO. We apply standard DPO to the same set of preference pairs $D_0^{\text{pairs}}$ as used in the first iteration of our method. Whether initializing from Llama-2-70b-chat ($M_0$) or from SFT training on the chosen (winner) examples, we find that DPO performance, while

---

[3]We have confirmed that the licenses of the datasets used in this paper (MIT for GSM8K and MATH, CC BY-SA 4.0 for ARC) are respected.

[4]If after filtering the number of pairs is larger than 60k, then we randomly select around 60k examples. This number is fixed because we do not want to introduce another source of variability in our experiments.

Table 1: **GSM8K results** comparing Iterative Reasoning Preference Optimization (Iterative RPO) against other baselines that are based on the same base model and training data. We report the exact match accuracy from a single generation (using greedy decoding), as well as majority voting over 32 generations (through sampling with temperature 0.8).

| Model | Test Accuracy (%) |
|---|---|
| Iterative RPO *(initialized from Llama-2-70b-chat)* | |
|    *Iteration 1* | 73.1 |
|    *Iteration 2* | 78.0 |
|    *Iteration 3* | 81.1 |
|      *w/ majority voting using 32 samples* | 88.2 |
|    *Iteration 4* | 81.6 |
|      *w/ majority voting using 32 samples* | 88.7 |
| *Other Llama-2-70b-chat-initialized methods* | |
|   Zero-shot CoT | 55.6 |
|      *w/ majority voting using 32 samples* | 70.7 |
|   DPO *initialized from Llama-2-70b-chat* | 61.8 |
|   DPO *initialized from SFT trained on Iteration 1 chosen seqs* | 60.3 |
|   SFT *on gold CoT examples* | 63.5 |
|   STaR (1 iteration) | 65.2 |
|   STaR (1 iteration, *but on twice as much data*) | 66.9 |
|   Iterative RPO (1 iteration, *but initialized from SFT trained on chosen seqs*) | 73.1 |
|   Iterative RPO (1 iteration, *but on twice as much data*) | 74.8 |

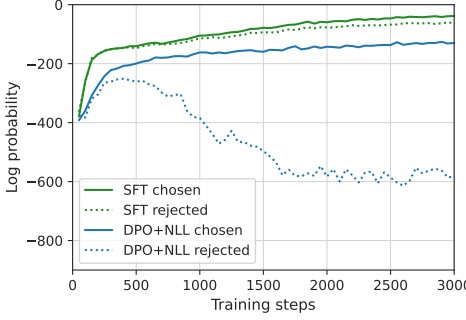

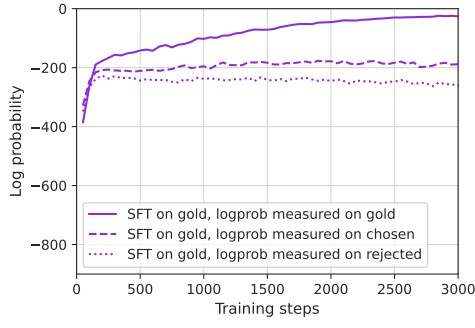

(a) SFT trained on chosen seqs; init from Llama     (b) SFT trained on gold CoTs; init from Llama

Figure 2: **Effect of SFT training**. **(a)** Although SFT training (solid green) is on chosen sequences ($D_0^{\text{pairs}}$, from iterative RPO iteration 1) only, the rejected sequence log probabilities (dotted green) also increase and are close to the chosen sequence probabilities. In contrast, our DPO+NLL training (blue) manages to decrease the rejected probabilities while increasing the chosen probabilities. This observation could potentially help explain why SFT-only performance lags significantly behind Iterative RPO Iteration 1 performance. **(b)** We show a similar plot but where SFT is trained on gold (dataset-provided) CoTs. Chosen and rejected sequence probabilities (which are from $D_0^{\text{pairs}}$) are still close to each other, but with a slightly bigger gap. Another observation is that the chosen sequence probabilities barely increase.

being better than zero-shot CoT, is no better than the SFT model, with accuracies of 61.8% or 60.3% respectively.

We also show that SFT on only the chosen CoT solutions, which corresponds to the first iteration of the STaR method, improves results to 65.2% over SFT on the gold solutions alone, but still falls short of the performance of the first iteration of Iterative RPO. One hypothesis for these improvements is the necessity of including the rejected sequences in the training objective; otherwise their probability

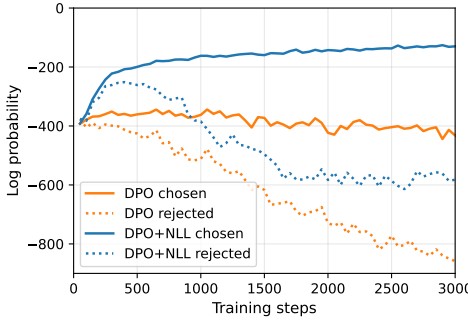
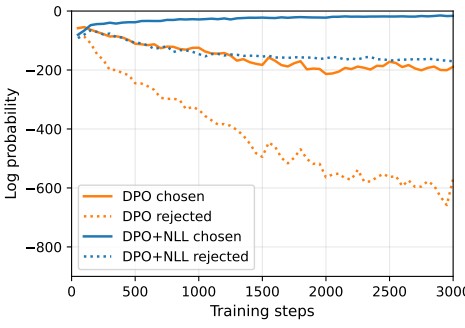

(a) Initialized from Llama          (b) Initialized from SFT trained on chosen seqs

Figure 3: **Effect of NLL loss term on DPO training for GSM8K**. In our GSM8K experiments we observe the log probability of chosen sequences in standard DPO without NLL loss (solid orange) decreases over training steps, especially if the model is initialized from SFT training on chosen sequences (right). However, they *increase* over training steps when using DPO with NLL loss (solid blue). In all four settings, the margin between the two curves continues increasing. We find that DPO+NLL loss gives superior test accuracy in our experiments.

increases along with the chosen samples; see Figure 2. We note this observation has also been reported in concurrent work [Hong et al., 2024].

All of the results reported above are using a single generation at test time using greedy decoding. If we use majority voting over 32 samples (sampling with temperature 0.8), a standard approach to improve performance in the literature, we can improve the accuracy of our approach from 81.1% to 88.2% for iteration 3, and from 81.6% to 88.7% for iteration 4 of Iterative RPO. While performance is much improved using majority vote, this should be compared to a majority vote baseline, where we find a similarly large improvement over the zero-shot chain-of-thought with majority vote, which obtains an accuracy of 70.7%.

**Iterations of Iterative RPO yield improved reasoning.** We observe that Iterative RPO provides improvements over its training iterations, increasing the base model accuracy by 47% (from 55.6% to 81.6%) in total. In contrast, supervised training using the gold CoT only brings about a 14% accuracy boost. We see performance improves across each iteration, from 73.1% to 78.0% to 81.1% to 81.6%. However, the gain decays across the iterations (17.5%, 4.9%, 3.1%, 0.5%), indicating an upper limit on learning across iterations, especially as we are iterating across a fixed number of prompts, i.e., only from the training samples.

We also show that it is the iterations of updating the model (i.e., initializing from the previous model) that are helping, not just because there is more data in the form of new pairs generated from the fixed training set. To test this statement, we run the first iteration of Iterative RPO but on twice as much paired data by doubling $K$, and we run the STaR method first iteration with twice as much data as well.[5] In both cases performance improves compared to less data, but not as much as performing two iterations. Iterative RPO with twice as much data obtains 74.8% (an improvement over 73.1% using the original dataset size); however, training for two iterations obtains 78.0%. For STaR, training on twice as much data obtains 66.9%, compared to 65.2% with the original data, which is still a much lower performance than Iterative RPO.

**NLL loss is necessary in our method: DPO with NLL vs. DPO without NLL.** The first iteration of our method can be compared to standard DPO training, which uses the same preference data, as reported in Table 1. We see a large performance drop (73.1% vs. 61.8%) using DPO compared to our method after one iteration. The gap remains large even when the standard DPO training starts from the superior SFT-tuned model, which it has been argued improves DPO's performance [Rafailov et al., 2023, 2024]. Our results support the need of the NLL loss term in our training, not just using SFT for initialization. To further understand this result, we plot the sequence-level log probability

---

[5]In particular, for "twice as much data" in Table 1, we generated more synthetic responses, so examples are not simple duplication.

over training steps for these methods in Figure 3. We see that for DPO without NLL loss there is a decrease over training for the chosen sequences, whereas for DPO with NLL there is not, which may help explain the improved performance of the latter. Related observations have been made elsewhere in various settings [Pal et al., 2024, Xu et al., 2024, Hong et al., 2024]. Further, we note that whether we initialize with Llama-2-70b-chat or SFT on chosen for Iterative RPO, accuracy results of first iteration training do not seem to deviate (both obtain the same score 73.1%). This is another advantage of our method as the training process is simpler without the SFT step.

**Other results in the literature.** We can compare our results to others in the literature, even if their experiments are in different settings. Touvron et al. [2023] reports an accuracy of 56.8% for 8-shot Llama-2-70b, which is close to our zero-shot CoT results for Llama-2-70b-chat. In terms of closed-source proprietary language models, some results are superior to ours, while others are not; for example GPT-4 obtains 92.0% (5-shot chain-of-thought) [Achiam et al., 2023], Claude 2 obtains 88.0% [Anthropic Team, 2023], PaLM 2 obtains 80.7% [Anil et al., 2023], while GPT-3.5 obtains 57.1% (5-shot) [Achiam et al., 2023]. We note that the size (number of parameters) and the makeup of the training set of some of these models have not been fully disclosed. For results that use the same size and class model, Llama-2-70b, MetaMath [Yu et al., 2024] reports an accuracy of 82.3%, while WizardMath reports 81.6% [Luo et al., 2023]. These last two results use additional augmented training data, whereas our method does not use additional prompts. Such approaches should be orthogonal to ours, and both can provide benefits.

## 3.2 ARC-Challenge Task

To test reasoning capabilities outside of mathematics, we employ ARC [Clark et al., 2018] which covers multiple science subjects. Questions are multiple-choice, for example: *"A fold observed in layers of sedimentary rock most likely resulted from"* with four possible answers, e.g., *"(A) cooling of flowing magma, (B) converging of crustal plates, (C) deposition of river sediments, or (D) solution of carbonate minerals"*. The training dataset contains 7.7k questions split into easy and challenge sets. We report results on the ARC-Challenge test set which has 1172 examples. There is no gold chain-of-thought reasoning provided for training examples in this task. Our method does not have that requirement and hence can still be applied as we only compute rewards based on the final answer. One consequence however is that if there is no model-generated correct solution for a question, then that question is not included in our training. We follow the same setup as before to first generate reasoning and then a final answer by the models (see Appendix B.2 for prompt) to construct data for iterations of Iterative RPO. We only train on the training set (both easy and challenge sets) and *do not* utilize the supporting ARC Corpus.

Specifically, in each iteration, we generate $N = 30$ solutions per problem using sampling with temperature 0.8 for iterations 1–2 and temperature 1.3 for iteration 3. We select $K = 20$ pairs of solutions per problem. We end up with around 20k example pairs for iteration 1, 11k example pairs for iteration 2, and 5k example pairs for iteration 3. The decrease in the number of examples is due to the lack of incorrect samples for a number of questions in later iterations. Each iteration is trained on a maximum of 4000 steps. The hyperparameter tuning relies on the provided development set.

We hence perform experiments using a very similar setup to the one previously described for GSM8K. Overall results are given in Table 2. We again find that Iterative RPO provides increased performance across iterations (84.8%, 86.2%, 86.7%) over three iterations. Majority voting using the model in the third iteration (32 samples, temperature 0.8) leads to another small boost (87.9%). These results outperform zero-shot CoT (77.8%), SFT on chosen sequences (79.8%) and standard DPO (83.5%). We arrive at similar observations in Figure 4a compared to Figure 3: when training with DPO without NLL loss, the log probabilities of chosen sequences barely increase over training; when training with DPO with NLL loss, the log probabilities increase noticeably.

Even though we arrive at similar conclusions to the ones from GSM8K, we find these results especially noteworthy due to the multiple-choice nature of the task. As there are typically only four possible answers, the generated data in step (i) of Iterative RPO may provide a CoT and a final answer that is correct by luck (as random guessing is correct 25% of the time). Hence, the nature of the task may introduce a significant amount of noise in the CoT generations used in preference optimization in step (ii). Nevertheless, the method seems robust to this issue and we still observe performance gains.

Table 2: **ARC and MATH results.** We compare Iterative Reasoning Preference Optimization (Iterative RPO) against other baselines that are based on the same base model and training data.

| Model | ARC-Challenge (0-shot) Test Accuracy (%) | MATH (4-shot) Test Accuracy (%) |
|---|---|---|
| Iterative RPO *(initialized from Llama-2-70b-chat)* | | |
| *Iteration 1* | 84.8 | 17.7 |
| *Iteration 2* | 86.2 | 19.9 |
| *Iteration 3* | 86.7 | 20.8 |
| *w/ majority voting using 32 samples* | 87.9 | 29.1 |
| *Other Llama-2-70b-chat-initialized methods* | | |
| CoT | 77.8 | 12.5 |
| *w/ majority voting using 32 samples* | 82.9 | 18.8 |
| SFT *on chosen sequences* | 79.8 | 16.8 |
| DPO *initialized from Llama-2-70b-chat* | 82.8 | 12.4 |
| DPO *init from SFT model trained on chosen seqs* | 83.5 | 10.5 |

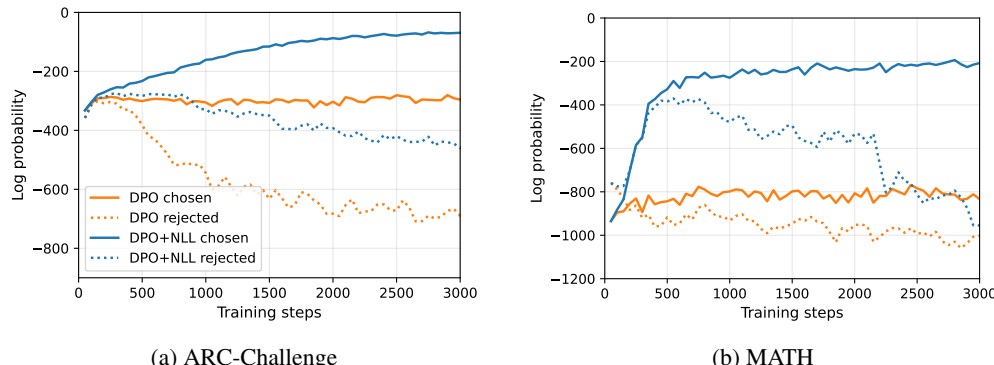

(a) ARC-Challenge  (b) MATH

Figure 4: **Effect of NLL loss term on DPO training for ARC and MATH**. The legend on the right plot is omitted due to space constraint, but it is the same as the legend in the left plot. Similar to GSM8K, in ARC-Challenge and MATH, we see that the log probabilities of chosen sequences barely increase over training steps when training with DPO. However, when training with DPO with NLL loss, the log probabilities increase over training steps.

## 3.3 MATH Task

We also experiment with more advanced math problems using the MATH [Hendrycks et al., 2021] dataset that is composed of 12,500 competition problems, for example the question: *"Tom has a red marble, a green marble, a blue marble, and three identical yellow marbles. How many different groups of two marbles can Tom choose?"*. While this may look superficially similar to the GSM8K task, it features substantially harder questions, as will be shown by the baseline performance. The test set has 5,000 examples. Similar to the GSM8K dataset, a gold CoT solution is provided for each problem, and the gold answers can be matched uniquely to predicted answers after normalization to compute rewards. We *do not use the accompanying pretraining data*. For each MATH question, we use a few-shot prompt given in Appendix B.2 as the input to the language model. In particular, the prompt includes four fixed in-context examples chosen from the training set. The language model needs these demonstrations so that the final answers can be properly formatted in LaTeX.

In each iteration, we generate $N = 20$ solutions per problem using sampling with temperature 0.8 for iterations 1–2 and temperature 1.0 for iteration 3. We select $K = 15$ pairs of solutions per problem, and after filtering out pairs with overly long generations, for each iteration we end up with around 75k example pairs. We train a maximum of 5000 steps per iteration; other details are similar to GSM8K setups.

Results are given in Table 2. We again find that Iterative RPO provides increased performance across iterations, from 17.7% to 19.9% to 20.8% over three iterations. Majority voting (32 samples, temperature 0.8) leads to a significant boost in performance (29.1%). These results outperform few-shot CoT (12.5%), SFT on chosen sequences (16.8%) and standard DPO (12.4%). In particular, DPO degrades the performance compared to initialization. Similar to the previous tasks, we show the log-probabilities during training in Figure 4b.

Overall, we find on all three distinct tasks we tried, from simpler to more difficult, similar observations on performance gains are exhibited by our method.

## 4 Related Work

**General iterative alignment methods.** Several works have implemented iterative reinforcement learning from human feedback (RLHF) with a human-in-the-loop to provide additional labels to retrain the reward model at each iteration, e.g., via Proximal Policy Optimization (PPO) [Schulman et al., 2017], reporting improvements across iterations [Bai et al., 2022, Touvron et al., 2023]. Recently, approaches have been proposed to perform iterative alignment without a human-in-the-loop. Iterative DPO [Xu et al., 2023, Xiong et al., 2023] optimizes preference pairs using DPO [Rafailov et al., 2023] at each iteration, and then constructs new preference pairs for the next iteration by generating them using the updated model, and scoring them using a reward model. Other iterative methods than DPO exist as well, such as the Cringe loss [Adolphs et al., 2023], Pairwise Cringe Loss [Xu et al., 2023], and ReST [Gulcehre et al., 2023].

SPIN [Chen et al., 2024] is an Iterative DPO-like framework that uses human labels as the winning response in a pair, and the last iteration's generations as the losing response in the pair. The authors note this has the limitation that once the model generations reach human performance, they are bottlenecked. Further, each input prompt is required to have a human-annotated generation. In contrast, our work only requires the final answer, but not the reasoning steps, and crucially uses the model to generate both winning and losing Chain-of-Thoughts. Only modest gains on reasoning tasks are reported in their work.

Self-Rewarding LLMs [Yuan et al., 2024] also use Iterative DPO with the LLM itself used as a reward model to construct pairs for each successive iteration. Both that work and the work of Rosset et al. [2024] and Snorkel AI Team [2023], which do similar iterations but with external reward models, show significant gains on general instruction following tasks. However, again, only modest gains on reasoning tasks are reported.

**Methods improving reasoning ability.** While a number of approaches have been developed to curate or distill training data for reasoning tasks [Yu et al., 2024, Toshniwal et al., 2024], in this work we focus on learning algorithms which is an orthogonal axis. Expert Iteration assumes a reward model, and repeatedly uses rejection sampling to filter generations and train on them, which is found to match the sample complexity of PPO [Havrilla et al., 2024]. STaR [Zelikman et al., 2022] relies on a similar loop: generate rationales to answer many questions, prompted with a few rationale examples; if the generated answers are wrong, try again to generate a rationale given the correct answer; and then fine-tune on all the rationales that ultimately yielded correct answers; and repeat. ReST$^{EM}$ [Singh et al., 2024] assumes a ground truth verifier and also fine-tunes on filtered samples in a repeated fashion. All these methods rely on finding high-quality samples for SFT-like training, rather than using DPO-like pairwise preference optimization as in our work.

The V-STaR method [Hosseini et al., 2024] trains a verifier using DPO and uses this to filter the generations of a model trained by SFT, rather than using DPO to train the generator, as we do. MAPO [She et al., 2024] also recently utilizes DPO but for multilingual reasoning tasks, where they translate across languages.

## 5 Conclusion

We propose an iterative training algorithm, Iterative Reasoning Preference Optimization, for improving chain-of-thought-based reasoning task performance in LLMs. In each iteration, we generate multiple responses and build preference pairs based on the correctness of their final answers, and then use a modified DPO loss with an additional NLL term for training. Our method does not

require human-in-the-loop or extra training data, and remains simple and efficient to implement. The experimental results show large improvements on GMS8K, MATH, and ARC-Challenge over various baselines using the same base model and training data. These results indicate the effectiveness of our recipe of iterative training in improving the reasoning capabilities of LLMs.

## Acknowledgments

We thank colleagues at Meta and NYU for valuable discussion: in particular, Angelica Chen, Jing Xu, Abulhair Saparov, Vishakh Padmakumar, Nicholas Lourie, Nitish Joshi, Ilia Kulikov, and J. Mark Hou.

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

# A    Limitations

On experiments: When training iteration $t$ using iterative RPO, we do not make use of the collected data in *previous* iterations. Utilizing those data could potentially boost the performance even more. We leave this point to future work, as it is not central to our theses. In addition, we have experimented on three tasks. It is unclear how the approach would perform on general instruction tuning tasks without a clear *best* answer, but we argue that positive results on the three tasks in this paper can already prove the method useful. The current recipe requires correct answers, and a clear metric for comparing a generated response with this correct answer.

Regarding our loss function: The NLL loss is shown to be helpful in our case. Our iterative RPO algorithm requires training data to be mostly collected from the previous iteration of the model. Therefore, the chosen and rejected sequences all have reasonably high probability under the model distribution. When training sequences are arbitrary (e.g., sampled from other models), it is unclear whether the NLL loss is necessary (although this setting does not fall under the umbrella of the iterative RPO procedure in this paper).

# B    More Details on Experimental Setup

## B.1    More Details on Hyperparameters

For baseline hyperparameter selection: We conducted a grid search for SFT, iteration 1 of DPO, and iteration 1 of RPO. For iteration>1, we use the same hyperparameters as iteration 1, except for the number of training steps which is selected individually for each iteration. For all baselines, learning rates are all tuned from the range of 5e-7 to 5e-6. DPO and RPO are tuned in the same sets of hyperparameters (if the hyperparameter exists in both methods).

## B.2    Prompts

**GSM8K.**    For each GSM8K question, we use the following prompt as the input to the language model:

> Your task is to answer the question below. Give step by step reasoning before you answer, and when you're ready to answer, please use the format "Final answer: ..."
> Question: [question here]
> Solution:

**MATH.**    For each MATH question, we use the following prompt as the input to the language model. In particular, the prompt includes four fixed in-context examples chosen from the training set of MATH. The language model needs these demonstrations so that the final answers can be properly formatted in LaTeX.

> Your task is to answer the last question below. Give step by step reasoning before you answer, and when you're ready to answer, please wrap your answer in \boxed, and conclude using the format "Final answer: ..."
>
> Question: [question for the first example]
> Solution: [solution for the first example]
> Final answer: [answer (e.g., number, formula) here]
>
> Question: [question for the second example]
> Solution: [solution for the second example]
> Final answer: [answer here]
>
> Question: [question for the third example]
> Solution: [solution for the third example]
> Final answer: [answer here]
>
> Question: [question for the fourth example]
> Solution: [solution for the fourth example]
> Final answer: [answer here]

Question: [the question to be solved]
Solution:

**ARC.**  For each ARC question, we use the following prompt as the input to the language model, assuming the question has four options (each question has three to five options).

Your task is to answer the question below. Give step by step reasoning before you answer, and when you're ready to answer, conclude using the format "Final answer: (insert letter here)"

Question: [question here]

(A) [option A here]

(B) [option B here]

(C) [option C here]

(D) [option D here]

Solution:

