# OpenReview forum: "Iterative Reasoning Preference Optimization"
_NeurIPS.cc/2024/Conference — NeurIPS 2024 poster_

### Official Review · Reviewer_DMPg · 2024-06-20

**Soundness:** 3
**Presentation:** 3
**Contribution:** 3
**Rating:** 6
**Confidence:** 4

**Summary:**

This work proposes an iterative training algorithm that enhances a model's Chain-of-Thought (COT) capabilities in reasoning tasks by combining self-improvement with preference optimization. The algorithm employs ground truth labels as supervision signals to evaluate model-generated responses, which are then incorporated into subsequent training iterations. Experiments on the GSM8K, MATH, and ARC benchmarks show significant improvements compared to the baseline, with continuous enhancement observed over multiple iterations.

**Strengths:**

This paper introduces preference optimization into the self-improvement of reasoning tasks, yielding excellent results.

**Weaknesses:**

1. In the experiments, this work conducts experiments on the training sets of the GSM8K, MATH, and ARC benchmarks, and evaluates models on the corresponding test sets. All tests were completed on held-in data, with no performance results provided for the model on held-out tasks.
2. This paper emphasizes the importance of preference optimization, distinguishing it from other iterative training methods like Rest-EM [1], which should be considered as a baseline.


[1] Singh, Avi, et al. "Beyond human data: Scaling self-training for problem-solving with language models." arXiv preprint arXiv:2312.06585 (2023).

**Questions:**

Please refer to the weaknesses section.

**Limitations:**

--

---

> ### Author Rebuttal · Authors · 2024-08-06
>
> We thank the reviewer for the comments.
>
>
> Weakness 1: Correct; we trained using IRPO by leveraging GSM8K/MATH/ARC training set, and we tested on the respective test sets, but not other datasets. This paper doesn’t focus on generalization to other datasets, but the reviewer raises a valid point. We believe that generalizing to other datasets might need a larger number of prompts (e.g., by leveraging a dozen different datasets for IRPO training) and it would be an effort to find good-quality test data as well.
>
>
> Weakness 2: We were not aware of this paper at the time of writing, and we will cite this paper. But as explained below, in fact, this algorithm is already considered in our baseline.
>
> This paper is similar to STaR (Zelikman et al.) from 2022. Our submission includes a variant of STaR baseline that uses temperature sampling without rationalization (i.e., without providing the correct answer first before generating the problem). In Rest-EM, in the binary reward case, when the solution has the wrong final answer, r=0; in this case we do not take any gradient. Therefore, the Rest-EM’s approach is essentially our variant of STaR. We’ll clarify this piece of related literature and our baselines in our revision, and thank you for suggesting!

---

> > ### Comment · Reviewer_DMPg · 2024-08-12
> >
> > Thank you for your response.

---

### Official Review · Reviewer_E9gS · 2024-07-11

**Soundness:** 2
**Presentation:** 3
**Contribution:** 2
**Rating:** 5
**Confidence:** 3

**Summary:**

The authors propose iterative RPO method for reasoning tasks. In particular, iteratively, the model at hand will be prompted to generate many CoT reasoning, and the ones that align with the true answers will be used as chosen, the other generated samples as rejected, for a DPO+NLL loss. The authors conduct experiments on GSM8K, ARC, and MATH to demonstrate the superiority of the proposed algorithm.

**Strengths:**

The reasoning task at hand is important to the community, and the proposed algorithm shows promising improvements.

**Weaknesses:**

The novelty between the proposed algorithm and self-rewarding language models seems to be marginal. It also occurs to me that more ablation studies should be conducted, see **questions** for detail.

**Questions:**

1. My main concern is where the improvement comes from. In iterative RPO, there are (at least) three sources of improvements (a) more data, and (b) the model is updated iteratively, and (c) preference optimization is used in addition to SFT.
    -  More data. While in table 1, iterative RPO with twice as much data is compared to STaR. It seems that the data is simply duplicated, but what if we use extra data generated by the model? Currently we are simply training more epochs.
    -  What if we take the positive and negative samples generated by iterative RPO, and used it to train DPO or SFT? This tells us how much (b) or (c) helps, respectively.
    - Currently, is the DPO baseline trained on just (x, y) or (c, x, y)?

2. A less important question is the importance of NLL. My intuition on using NLL is that it regularizes the model from hacking the reward too much, but conceptually, choosing a larger $\beta$ does the same thing -- it enforces a larger KL regularization between the SFT model and the trained model. If you vary $\beta$, will figure 2 still hold?

**Limitations:**

Limitations have been discussed.

---

> ### Author Rebuttal · Authors · 2024-08-06
>
> We thank the reviewer for the feedback and insights.
>
> Re: novelty.
> - Our method is not self-rewarding because the reward signals come from the ground-truth labels instead of the model itself. More specifically, self-rewarding LM requires generating prompts, and using LMs to evaluate the sampled generations. But LMs are horrible at evaluating reasoning-related generations (e.g., according to RewardBench – although there has been some progress in the past few weeks). So we believe that self-rewarding LMs cannot be successfully directly applied using llama-2-70b-chat, but using IRPO in an unsupervised fashion is a really promising research avenue!
> - We are proposing a loss that integrates NLL while the self-rewarding LM uses standard DPO training. Through our experiments (in this IRPO submission), we demonstrate the importance of the NLL term on positive examples.
> - The focus of self-rewarding was general instruction tuning and our paper focuses on reasoning, math, and science reasoning in particular. In fact, the self-rewarding did not bring improvement in the math category.
>
> Q1:
> - IMPORTANT CLARIFICATION on “more data”: The “twice as much data” baseline means that we actually *generated* twice as much data – there is no duplication. We apologize for the confusion and will emphasize this point in the paper.
> - On the second bullet point: “Using the positive & negative samples for DPO” is actually our DPO baselines in the tables. “Using positive samples for SFT” is actually the STaR baseline in the tables.
> - On the third bullet point: (c, x, y). The DPO baseline is essentially RPO but without the NLL term.
>
> Q2: Yes. We experimented with different beta’s (0.05, 0.1, 0.5, 1.0) and the probability trends are similar within each dataset. We are investigating why sometimes naive DPO leads to decreasing chosen AND rejected probabilities like in Figure 2. Our current hypothesis is that it’s related to the fact that both current and rejected generations are sampled from the most recent model – both generations have high probabilities under the most recent model distribution.

---

> > ### Comment · Reviewer_E9gS · 2024-08-12
> >
> > Thanks for the response. With regard to Q2 second point, I meant train DPO and STaR on the data generated by 4 iterations of Iterative RPO and see how more generated data can help. Meanwhile, since Iterative RPO also incorporates NLL Loss, for fair comparison, the DPO baseline should also incorporate NLL so that we can understand where the improvement comes from.

---

> > > ### Author Response · Authors · 2024-08-13
> > >
> > > To reviewer E9gS:
> > >
> > > Thank you for your reply!
> > >
> > > Do you mean the second point of Q1 (instead of Q2)?
> > > - When training for M2, we’ve tried using iteration 1 data and iteration 2 data **together** (rather than our proposed method of iteration 2 data only, but initialized from M1). The result for gsm8k is actually worse than **only** using iteration 2 data initialized from the iteration 1 model M1 (72.1 instead of 73.1). This indicates that **the data quality is more important than the amount**.
> > > - Using data from iterations necessarily requires iterations in the first place. Collecting 4 iterations of data requires IRPO training iterations in the first place, so even if training on that data works we cannot conclude that iterations are unnecessary.
> > >
> > > If the reviewer is wondering where the improvement comes from, it comes from 2 sources:
> > > - **It comes from the added NLL term**, because our approach outperforms the model trained without the NLL term (regular DPO) in the first iteration.
> > > - **It comes from iterations**, because more generated data (from the current model) don’t help as much. 2x data (no duplicates) doesn’t work as well – see last two rows of Table 1. In contrast, training on higher quality data generated from the improved model from the last iteration brings better performance.
> > >
> > > These two are the sources of improvement.

---

> > > > ### Comment · Reviewer_E9gS · 2024-08-14
> > > >
> > > > I appreciate the responses from the authors. My concerns are addressed. Since I still believe that the technical novelty on top of self-rewarding LLM is marginal, I'm raising my score to 5.

---

### Official Review · Reviewer_9NDK · 2024-07-13

**Soundness:** 2
**Presentation:** 3
**Contribution:** 3
**Rating:** 5
**Confidence:** 3

**Summary:**

This paper proposes a novel method for preference optimization for reasoning tasks. Their method involves prompting models to generate the CoT reasoning steps and answers for a set of reasoning task inputs, then labeling samples as correct or incorrect using the ground truth outputs, and training the preference model. The authors use standard DPO loss with the addition of a negative sampling loss using samples for which the model generates incorrect outputs. They train models iteratively, using the previous generation as the base model to generate new outputs at each step. They observe that using this method they are able to improve significantly over DPO for three reasoning tasks.

**Strengths:**

RPO demonstrates performance boosts on a variety of tasks, outperforming DPO and SFT. Improving model performance for reasoning tasks has shown to be a challenging task, and this method shows promise for improving model performance in this area.

**Weaknesses:**

1. Although the authors present ablations on the number of data points for RPO, they do not present experiments with simply training the model for longer. This is true for the comparisons to DPO as well. Though iterative training likely is playing a role in the performance of the model, for the sake of careful analysis, it would be good to compare the performance of a model trained for multiple generations vs a model trained for only one generation but an equivalent number of steps.

2. I was not able to find details on the hyperparameters used for DPO or SFT or how optimized they were. In contrast, the hyperparameters for RPO are chosen carefully. While it's unlikely all performance boosts are due to this, it should be clear how hyperparameters were chosen for the sake of comparison and reproducibility.

3.  Though the comparison of DPO with and without NLL shows that the loss for rejected samples decreases when NLL is applied, it does not demonstrate why this is important for model improvement. A more effective analysis would compare the mistakes made by the model with and without this loss (e.g. FNR/TNR).

**Questions:**

1. What hyperparameters are used for DPO and SFT? Are they tuned with the same care as those for RPO?

2. Do you explore the reasoning steps generated by models? How often are they actually correct?

3. What is the cost of training models using this method?

**Limitations:**

The authors address limitations in the appendix.

---

> ### Author Rebuttal · Authors · 2024-08-06
>
> Thank you for the detailed review and suggestions!
>
>
> **Weakness 1 (longer training)**: Thank you for pointing out this issue. In each iteration, we do train longer (5000 steps) but end up selecting an earlier checkpoint – the selected checkpoints (by validation accuracy) are usually trained for 2000 or 3000 steps (e.g., for GSM8K, M1 is trained for 2000 steps and M2 for 2000 steps). This is because training longer makes the model overfit to the training and hurts validation performance. We use the same checkpoint selection process for both RPO and DPO. We’ll include more detailed discussion in the revision.
>
> Importantly, in Table 1 of the submission, we also included results where the model is trained on twice as much data instead of two iterations (using STaR and using iterative RPO, also for a max of 5000 steps). As shown in Table 1, this strategy (the last row) doesn’t match the performance of doing two separate iterations (2nd row). This highlights the advantage of iterations over longer training.
>
> Another piece of evidence (that iterative training is helpful) is that recent work (Preference Fine-Tuning of LLMs Should Leverage Suboptimal, On-Policy Data; https://arxiv.org/abs/2404.14367) has demonstrated the importance for on-policy data (i.e., iterations). We’ll make sure to discuss this issue more thoroughly.
>
>
> **Weakness 2 (hyperparameters for baselines)**: We conducted a grid search for SFT, iteration 1 of DPO, and iteration 1 of RPO. For iteration>1, we use the same hyperparameter as iteration=1 (except for num_training_step which is selected individually for each iteration). All experiments share a similar config file; therefore, for all baselines, learning rates are all tuned from the range of 5e-7 to 5e-6. DPO and RPO are tuned in the same sets of hyperparameters (if the hyperparameter exists in both methods). We will clarify this issue, and thanks for pointing this out.
>
>
> **Weakness 3**: With DPO training, the probability of rejected samples decreases regardless of NLL (see Fig3 a). Addition of NLL affects the chosen samples and makes them go up in probability. In contrast, their probability doesn’t go up much in DPO training without NLL. The chosen samples are correct solutions, so their higher probability means the model is likely to generate that correct solution. The direct way to measure if the model is generating correct solutions or not is to look at the test set accuracy, which we report in the paper.
>
> In other words, our hypothesis is that without NLL, both chosen and rejected probabilities decrease, and given that probabilities sum to one over sequences, the probabilities might be going to unforeseen low-quality sequences.
>
>
> Q1: Yes. Please see weakness 2 above. Thanks again for raising this point.
>
> Q2: We have eyeballed ~30 generations for each dataset; to the best of our ability (given that MATH is difficult), if the answer matches, then our generated CoT is almost always correct for these particular datasets, especially GSM8K and MATH.
>
> Q3: Generation is relatively cheap these days given the low-precision toolkits (e.g., vLLM; https://github.com/vllm-project/vllm) – using 8 V100 (32GB) GPUs, we can sample a few thousand responses (for our tasks) from 70B models in just a few minutes. All training in this paper is done using eight nodes, each containing eight A100 GPUs (briefly mentioned in Section 3); each 10 steps take around 2 minutes; training 5000 steps (plus validation and checkpoint saving) takes less than a day.

---

> > ### Comment · Reviewer_9NDK · 2024-08-13
> >
> > My apologies for the late reply and thank you for clarifying these points. The clarifications in this rebuttal and others have cleared up the main concerns I had, and I'm comfortable to increase my score slightly.

---

### Official Review · Reviewer_jeVs · 2024-07-13

**Soundness:** 4
**Presentation:** 4
**Contribution:** 4
**Rating:** 8
**Confidence:** 2

**Summary:**

The paper introduces a novel approach to improve the performance of language models on reasoning tasks through iterative preference optimization. It proposes an iterative method that generates multiple reasoning steps and final answers, constructs preference pairs based on the correctness of the answers, and then optimizes these pairs using a modified Direct Preference Optimization (DPO) loss combined with a negative log-likelihood (NLL) term. This iterative process results in progressively improved model performance. The approach demonstrates significant accuracy improvements on the GSM8K, MATH, and ARC-Challenge datasets using the Llama-2-70B-Chat model, outperforming other models that do not rely on additional datasets. Key contributions include the iterative application of DPO with an NLL term for reasoning tasks, comprehensive experimental validation, and performance gains without the need for additional human-annotated data.

**Strengths:**

1. The paper is well-written, and easy to follow.
2. The idea is clean and the contribution is clear.
3. The evaluation and improvements are convincing and significant.

**Weaknesses:**

I do not see significant weakness.

**Questions:**

Although the authors have discussed other related iterative preference optimization methods (e.x. Iterative DPO, Self-rewarding, SPIN), why not include their results in the evaluation tables?

**Limitations:**

See Questions.

---

> ### Author Rebuttal · Authors · 2024-08-06
>
> We thank the reviewer for the review and the support!
>
> We presented DPO results in the paper. We showed that DPO in iteration 1 does significantly worse than our RPO’s iteration 1; hence we did not try further iterations of standard DPO (aka Iterative DPO). The reviewer raises a good point that we can check what future iteration results are like. SPIN doesn’t include the NLL term in IRPO, so it’s closer to iterative DPO than IRPO; moreover, SPIN assumes always using the reference CoT but we do not; they also report only modest gains in reasoning tasks.
>
> The reviewer raises a good point about self-rewarding LM. The original algorithm requires generating prompts and evaluating the sampled responses using the LM itself. IRPO assumes knowing what the correct answer is (e.g., the answer may be 7.5 for a math question), but we wouldn’t know the correct answers for augmented prompts. LMs are quite bad at generating evaluations of a response (e.g., according to RewardBench – although there has been some progress in the past few weeks). So we believe that self-rewarding LM cannot be successfully directly applied using llama-2-70b-chat, but using IRPO in an unsupervised fashion is a really promising research avenue to keep exploring!

---

> > ### Comment · Reviewer_jeVs · 2024-08-13
> >
> > Thanks for your response.

---

### Decision · Program_Chairs · 2024-09-25

**Decision:**

Accept (poster)

**Comment:**

The paper presents a method that iteratively preference optimization based on the correctness of the generated answers. In other words, they optimize the preference between competing generated CoT candidates and this progressively contrastive appraoch leads to rapid gains when the model is already well-trained. The authors adopt a modified DPO objective with a NLL term. It lead to significant performance improvements while not using external data, boosting the GSM8k result from 55.6 to 88.7. Some reviewers had concerns about the sources of improvements and whether it comes from using more generated data. The concerns were addressed by the authors.